

# A Sub-Grid Parameterization Scheme for Topographic Vertical Motion in CAM5-SE

Yaqi Wang[1], Lanning Wang[1,2*], Juan Feng[1], Zhenya Song[3], Qizhong Wu[1], Huaqiong Cheng[1]

5   [1]College of Global Change and Earth System Science (GCESS), Beijing Normal University, Beijing 100875, China
[2]Laboratory for Regional Oceanography and Numerical Modeling, Pilot National Laboratory for Marine Science and Technology, Qingdao 266237, China
[3]First Institute of Oceanography, and Key Laboratory of Marine Science and
10  Numerical Modeling, Ministry of Natural Resources, Qingdao 266061, China

*Corresponding to: Lanning Wang (wangln@bnu.edu.cn)*


## Abstract:

Overestimation of precipitation over steep mountains is always a common bias of atmospheric general circulation models (AGCMs). One basic reason is the imperfection of parameterization scheme. Sub-grid topography has a non-negligible role in the dynamics of the actual atmosphere, and therefore the sub-grid topographic parameterization schemes have been the focus of model development. This study proposes a sub-grid parameterization scheme for topographic vertical motion in CAM5 to revise the original vertical velocity by adding the topographic vertical motion and then resulting a significant improvement of simulation in precipitation over steep mountains. The results show a better improvement in precipitation simulation in steep mountains, such as the steep edge of the Tibetan Plateau and the Andes. The positive deviations of the precipitation on the mountain tops and the negative deviations in the windward slope are revised. The improved scheme of topographic vertical motion reduces the model biases of summer mean precipitation simulations by up to 48% (6.23 mm day$^{-1}$) on the mountain tops. The improvement of convective precipitation (4.83 mm day$^{-1}$) contributes the most to the improvement of the total precipitation simulation. In addition, we extend the dynamic lifting effect of topography from the lowest layer to multiple layers, approaching the surface layer. Moreover, the water vapor transport in low-altitude regions in front of the windward slope is also considerably improved, leading to simulations of much more realistic circulation patterns in the multi-layer scheme. Since the sub-grid parameterization scheme addresses the more detailed problem caused by topography, the water vapor is transported further to the northwest in the multi-layer scheme. The topographic vertical motion schemes in both the Single- and Multi-experiments can improve the model performance in simulating precipitation in all regions with complex terrain.

## 1 Introduction

Numerical models have been widely used and become an essential tool to predict and simulate the weather and climate. However, there are still large deviations compared with observations, especially for precipitation simulation and prediction. It is of great scientific and social relevance to accurately simulate precipitation by using atmospheric general circulation models (AGCMs). In particular, the Coupled Model Intercomparison Project Phase 5 (CMIP5) and Phase 6 (CMIP6) models always overestimate the precipitation in regions with steep topography, which have been investigated in previous studies (Liu et al. 2014; Akinsanola et al. 2021; Cui et al. 2021). Jia et al. (2019) found that all CMIP5 models overestimate the monthly precipitation over the Tibetan Plateau by an average of 48.2 mm (~150%), with larger biases during spring and summer. Zhu and Yang (2020) also found that the systematic model biases (overestimation of precipitation) in the Tibetan Plateau still exist, even performed more poorly based on the performance of 23 models in CMIP6. Similar problems also exist in precipitation simulations in other mountain regions with steep terrains, such as the Andes in South America, the Rocky Mountains of North America,



and Indonesia. Excessive precipitation was simulated in both weather/climate models and global/regional models in regions with steep and high mountains, but less precipitation in windward slopes (Done et al., 2004; Kunz and Kottmeier, 2006; Alpert et al., 2012; Chao 2012; Navale and Singh, 2020).

The reasons for excessive precipitation simulated by numerical models over steep mountains are complex, involving the horizontal resolution, dynamical cores, physical processes, and their complicated interactions (Liang et al., 2021). There is plenty of evidence of a close relationship between orography and precipitation patterns at spatial scales of a few kilometers, even in climatological precipitation rates. Thus, improving model resolution is a possible way to improve the biases of precipitation simulations. Kimoto et al. (2005) found that higher-resolution versions of General Circulation Models (GCMs) can better characterize the frequency distributions of different precipitation patterns. Similar results can be found in regional models. Lin et al. (2018) compared the simulations with resolutions of 30 km, 10 km and 2 km based on the Weather Research and Forecasting model, and they found that higher-resolution simulations can reduce positive precipitation biases over the Tibetan Plateau. However, increasing spatial resolution does not always improve precipitation simulations, for example, in lowlands of southeastern England (Chan et al. 2013; Wang et al. 2017). The relationship between the spatial resolution of models and the quality of precipitation simulation remains elusive. Additionally, high-resolution climate models require a large amount of computation and storage. Some parameterization schemes are also proposed to improve the accuracy of precipitation simulation, which mainly focus on the parameterization schemes for physical processes. For example, in the past 20 years, much effort has been made to develop stochastic convection schemes and apply them to numerical models, resulting in some substantial improvements in precipitation simulation (Chen et al. 2010; Fonseca et al. 2015; Wang and Zhang, 2016; Attada et al. 2020).

The simulation bias of topographic precipitation has been a challenge for numerical models. Most studies are based on improving model resolution and the parameterization schemes of physical processes, but few studies focused on the modification of the dynamic core for numerical models, especially the dynamic lifting. At spatial scales greater than approximately 40 km and for mountain ranges exceeding approximately 1.5 km in height, the maximum condensation is generated over low, steep and windward slopes due to upslope flow (Roe 2005). An important quantity of orographic precipitation is water vapor flux. In numerical models, Yu et al. (2015) replaced the semi-Lagrangian method with a finite-difference approach for the trace transport algorithm to restrain the "overshoot" of water vapor to the high-altitude region of the windward slopes. Codron and Sadourny (2002) tested the advected water vapor with respect to saturation values and redistributed it accordingly over the grid points found along the advecting path. Actually, these two schemes add the limitation of oversaturation for water vapor advection, which may cause partial precipitation when the water vapor advects upward mountain slopes along terrain-following



coordinates. Less water vapor is transported to summits and plateaus and settles in windward slopes and foothills in advance, thus improving precipitation simulations in steep mountains. These studies only improve the scheme of water vapor advection scheme, and only Shen et al. (2007) proposed a sub-grid correction parameterization scheme for pressure tendency by considering slope and orientation according to the disturbance lifting caused by each fine grid. Based on this, the precipitation simulation in the P-$\sigma$ regional climate model of Nanjing University over complex terrain areas was improved. But it is only a case study of precipitation simulation in East China.

As mentioned above, sufficient water vapor and dynamic lifting are the necessary conditions for precipitation (Shen et al. 2021). Considering the shortcomings of the current dynamic lifting studies for numerical models, in this study, we propose a sub-grid parameterization scheme of topographic vertical motion and apply in CAM5, one of global atmosphere general circulation models, to improve precipitation simulation in areas with complex terrain. In particular, we extend the dynamic lifting effect of topography on airflow from the lowest model layer to multiple layers and consider the influence of the decay of vertical airflow.

The remainder of this paper is organized as follows. Section 2 describes the modeling context and the data used in this research and details the sub-grid parameterization scheme for topographic vertical velocity. Section3 analyzes and compares the precipitation simulated by two topographic vertical velocity experiments. The main conclusions and discussion are presented in section 4.

## 2 Model, methodology and experiments

### 2.1 CAM5-SE

The models used in this study are the Community Earth System Model (CESM; Hurrell et al. 2013) version 1.2.1. from the National Center for Atmospheric Research (NCAR) and the Community Atmospheric Model version 5 (CAM5; Neale et al. 2010) with the new spectral element dynamical core (CAM-SE). The CAM-SE is based on the High-Order Method Modeling Environment spectral element method (HOMME, Dennis et al. 2012) and adopts a conventional vector-invariant form of the moist primitive equations. Noted that the CAM-SE uses the vector-invariant form of the momentum equation instead of the vorticity-divergence equation. The pressure vertical velocity can be expressed by $\omega = D_p/D_t$, as shown in Eq. (1).

$$\omega = \frac{\partial p}{\partial t} + \vec{u} \cdot \nabla p + \dot{\eta}\frac{\partial p}{\partial \eta} = \vec{u} \cdot \nabla p - \int_{\eta_{\text{top}}}^{\eta} \nabla \cdot \left(\frac{\partial p}{\partial \eta}\vec{u}\right) d\eta', \qquad (1)$$





The major model physics of CAM5-SE include the separate deep convection scheme (Zhang and McFarlane 1995; Richter and Rasch 2008), the University of Washington shallow convection schemes (Park and Bretherton 2009) and a moist turbulence scheme (Gettelman et al. 2013) for calculating sub-grid vertical transport of heat and moisture. The cloud microphysics (Morrison and Gettelman 2008; Gettelman et al.

2010) includes both the direct and indirect effect for sulfate and black and organic carbon. For tracer advection, CAM-SE is modeled closely on the finite volume core. It uses the same conservation form of the transport equation and the same vertically Lagrangian discretization (Lin, 2004). The radiation scheme is Raipid Radiative Transfer Model for GCM (RRTMG) package (Mlawer et al. 1997).

## 2.2 Topographic vertical motion and sub-grid topography

## parameterization scheme

Alpert and Shafir (1989) found that orographic precipitation at micro/meso scales is highly predictable with the adiabatic assumption that the lifting is determined by $V \cdot \nabla Z_s$. The surface vertical velocity caused by the forced lifting of topography can be

expressed by Eq. (2).

$$\omega_s = \mathbf{V}_0 \cdot \nabla Z_s, \qquad (2)$$

In the P-coordinate system, Eq. (2) can be rewritten as Eq. (3):

$$\omega = \frac{dp_s}{dt} = \frac{\partial p_s}{\partial t} + \vec{V}_s \cdot \nabla p_s, \qquad (3)$$

Where $\vec{V}_s$ and $p_s$ indicate the surface wind velocity and the surface pressure, respectively. After considering the topographic vertical velocity, Eq. (3) can be rewritten as Eq. (4).

$$\omega = \omega_0 + \omega_s, \qquad (4)$$

$$\omega_s = -\rho g \vec{V} \cdot \nabla Z_s = -\rho g \cdot |\vec{V}| \cdot \tan\theta_N \cdot \cos(\theta - \varphi_N)$$

$$= -\rho g \sqrt[2]{u^2 + v^2} \cdot \tan\theta_N \cdot (\cos\theta \cdot \cos\varphi_N + \sin\theta \cdot \sin\varphi_N), \qquad (5)$$

where $\omega_s$ denotes the topographic vertical velocity of the lowest model layer, $\theta$ is the wind direction, $\theta_N$ is the slope, and $\varphi_N$ is the aspect, $\rho$ and $g$ are constants. It can be seen that the surface topographic vertical velocity is proportional to the surface





wind speed, the tangent of the slope and the cosine of the angle between the mountain aspect and the wind direction. Figure 1a shows the distribution of surface topographic vertical velocity with the slope and the angle between the wind direction and aspect under unit wind speed. In fact, the angle between the mountain aspect and the wind direction ranges from 0° to 360°. When the angle in the range of 0°–90° or 270°–360°, it indicates an ascending motion, while the angle of 90°–270°, it represents a descending motion. The angle range of 0°–90° is chosen just because it can cover the range of cosine values and is adequately representative. This study only focuses on the simulation of precipitation caused by blocking uplift in windward slopes. At the current model resolution, the maximum slope captured by Digital Elevation Model (DEM) data is 61°,, indicating that the maximum surface topographic vertical velocity is about 22Pa/s, and is positively correlated with slope. That is, when the mountain is the steepest and the angle between the wind direction and aspect is the smallest, the topographic vertical velocity reaches the maximum. However, when the slope is less than ~5°, the topographic vertical velocity is too small to be ignored.

Generally, only the single layer, that is the lowest layer, is considered in numerical models. However, in the actual atmospheric movement, the vertical motion not only affects the surface layer, but also affects near surface layers. Although it is realized in theory, it has not been applied in weather/climate prediction. Thus, we extend the vertical velocity from single layer to multi layers, as shown in Eq. (6):

$$\omega = \omega_0 + \omega_s \times \gamma, \qquad (6)$$

where $\gamma$ indicates the attenuation coefficient of topographic vertical velocity $\omega_s$ and it increases with the elevation, as shown in Eq. (7):

$$\gamma = \frac{sh(\sqrt[2]{2}\frac{2\pi}{L}\sqrt{\frac{\sigma}{f^2}} \times p)}{sh(\sqrt[2]{2}\frac{2\pi}{L}\sqrt{\frac{\sigma}{f^2}} \times p_0}, \qquad (7)$$

where $f$ represents the Coriolis term, $p_0$ is the reference pressure, $p$ is the actual pressure, $\sigma = -\frac{T}{\theta}\frac{\partial \theta}{\partial p}$ is a constant, and $L$ is the wavelength. Because the complexity of hyperbolic sine function calculation and the fact that the initial pressure in complex terrain areas actually does not start from the sea level but from the surface layer, we simplify Eq. (8) according to Taylor's formula to make $\gamma$ become an exponential function that varies only with latitude and pressure difference $\Delta p$:

$$\gamma \approx e^{\left(\frac{\sqrt{\sigma}}{2dl \times f \times \sin(lat)}\right) \times (-\Delta p)}, \qquad (8)$$





where $\Delta p$ indicates the difference between the surface pressure and the pressure on a certain model layer, $dl$ is model horizontal resolution. $\frac{\sqrt{\sigma}}{2dl \times f \times sin(lat)}$ is static variable which can be preprocessed at each integration step without calculation. After simplification, the divergence of $\gamma$ between Eq (7) and Eq (8) is only $10^{-10}$. Thus, the simplified Eq. (8) can be applied in numerical models to calculate the multi-layer topographic vertical velocity.

Figure 1b shows the linear variation of the unit topographic vertical velocity intensity with altitude at the given model resolution. The results indicated that with the increase of model resolution, the topographic vertical velocity decreases rapidly with altitude. When $L$=10km, that is high-resolution numerical models and $\Delta p$=10hPa, there is almost only one layer in the model, and the vertical velocity decreases to be negligible. In low-resolution numerical models ($L$=150 km), when $\Delta p \geq$ 150hPa, the influence of topography on the vertical motion can be negligible. Therefore, the influence on the attenuation of multi-layer topographic vertical velocity can be ignored in high-resolution numerical models but can be considered in low-resolution numerical models. It can provide some new information for numerical simulations. Notably, preprocessing the sub-grid topographic data before the model integration may increase a small amount of computation compared with CAM5-SE.

[Insert Figure 1]

The trigonometric function of slope and aspect calculated by Eq. (5) is parameterized to the model dynamic processes to evaluate the topographic vertical motion. A realistic statistical method based on trigonometric function transformation to calculate sub-grid slope and aspect for describing the orographic characteristics of complex areas over the globe has been proposed and tested in Wang et al. (2022). Since tan(slope)×cos(aspect) ( $tan\theta_N cos\varphi_N$ in Eq. (5)) and tan(slope)×sin(aspect) ($tan\theta_N sin\varphi_N$ in Eq (5)) conform to the Gaussian distribution, it is transformed into the standard normal distribution (standardized transformation) before processing.

According to the probability P table, the value of $Z_p$ corresponds to $Z$ during Z-Scores ( $Z = \frac{x-\mu}{\sigma}$, $\mu$ denotes the average, and σ indicates the standard deviation).

Then a representative value of several sub-grid topography values at the model grid scale is selected and can be easily described and applied. Before the experiments, advanced preprocessing is used to calculate the probability densities of the trigonometric function and grid weights.

## 2.3 Experimental design and data

The CAM stand-alone model can be run using CESM scripts, which is coupled to a



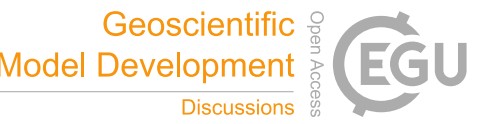

data ocean model, a thermodynamic sea ice model and an active land model, when one of "F" component sets of CESM is chosen. We choose the F_2000_CAM5
component set of CESM to conduct numerical experiments. The simulations are performed at the horizontal resolution of ne30 (about 1°) and 30 hybrid sigma-pressure levels, with an integration time step of 1800 s. Three 6-year simulations are forced by the prescribed current sea surface temperature and sea ice range with seasonal variations and are recycled yearly (Stone et al. 2018). The one
without any modification is the control experiment (Ctl experiment). The others are the sensitivity experiments, which are the same as the control experiment but consider the lowest topographic vertical velocity (Single experiment) and the decrease of multi-layer topographic vertical velocity (Multi experiment). All the three cases are carried out for 6 years, and the first year of simulation is discarded to avoid any
spin-up.

The topography data used is the United States Geological Survey (USGS) DEM data with the resolution of 1km×1km. The Global Precipitation Climatology Project (GPCP) Level 3 Monthly 0.5-Degree V3.0 beta (Huffman et al. 2019) from 1987 to 2016 is used to evaluate the simulated precipitation. Monthly mean atmospheric data,
comprising surface pressure, specific humidity, zonal and meridional wind ((at 11 vertical levels from 1000 to 700 hPa) during 1991–2021, are from the European Centre for Medium-Range Weather Forecasts Reanalysis 5 data set (ERA5) on a 0.25° × 0.25° grid (Hans et al., 2020).

## 2.4 Improvement or divergence ratio

Divergence ratio is an indicator used to measure the difference ratio between simulation results and observation results. Improvement ratio is an indicator used to measure the improvement ratio between Single (Multi) and Ctl experiments. In mountain meteorology, the precipitation enhancement ratio (PER) is the ratio of the precipitation $P$ at mountain peak or some other selected points to the precipitation at
the reference point or in the reference region $P_{REF}$, as presented in Eq. (9).

$$\mathrm{PER} = \frac{P}{P_{REF}}, \qquad (9).$$

The reference region should be far enough removed that it is unaffected by the mountain, but still in the same climate zone (Smith 2019). We extend Eq. (9) to any physical quantity to obtain Eq. (10).

$$\mathrm{PER} = \frac{\Delta P}{P_{REF}}, \qquad (10)$$

where $\Delta P$ indicates the difference in simulations between the sensitivity and control



experiments or the difference between the simulations from the control experiment
and observation data. $P_{\mathrm{REF}}$ represents simulations from the control experiment. Then,
the PER reflects the improvement ratio or divergence ratio.

## 3 Results

A region of 22°N–45°Nand 70°E–105°E is selected to cover the Tibetan Plateau. The

Tibetan Plateau is influenced by the plateau monsoon and has a distinct seasonal
pattern of wet summer and dry winter (Su et al. 2013). The precipitation reaches its
annual maximum in summer, accounting for 60%–70% of the annual accumulated
precipitation (Yanai and Wu 2006; Wang et al. 2018). Therefore, summer precipitation
is of great significance for this study in the region.

The geographical distributions of boreal summer (June–August, JJA) mean
precipitation amount from GPCP, Ctl, Single and Multi experiments are shown in Fig.
2. In summer, most precipitation over East Asia is related to the Indian summer
monsoon and the East Asian summer monsoon (Tao and Chen 1987). The results
indicate that for the GPCP (Fig. 2a), a large rainfall amount is concentrated in the Bay

of Bengal and the southeastern periphery of the Tibetan Plateau, but for the
simulations from the Ctl (Fig. 2b), Single (Fig. 2c) and Multi (Fig. 2d) experiments,
little rainfall is received in these areas. However, the precipitation increase appears on
the southern slope of the Tibetan Plateau in model experiments, but there is little
rainfall in this region in GPCP.

[Insert Figure 2]

In order to illustrate the biases of the model simulation and the improvement of the
topographic vertical motion scheme, the differences in the summer precipitation
between sensitivity experiments, Ctl experiment and GPCP are shown in Fig. 3. The
most striking feature of the bias distribution is its close relation with topography.

Positive precipitation bias controlling the Tibetan Plateau has been a common error in
many climate models for a long time (Yu et al., 2015). The largest overestimations of
the Ctl experiment (Fig. 3c) are found over the eastern and southern edges of the
Tibetan Plateau, mostly in the regions with altitudes of 500 m and 4000 m. According
to Eq. (10), the divergence ratio is about 80%(Fig.3f). In addition, the larger

underestimations of precipitation can be found in front of the southern slope of the
Tibetan Plateau, mostly in the region below the altitude of 500m. The region with the
largest underestimation is located in the area of 22°N, 90–98°E, with an
underestimation ratio of about 100%. However, underestimation ratios in other
regions are 20–40%. This result indicates that the southwesterly wind transports the

water vapor from the ocean to the southern slope of the Tibetan Plateau. Due to the
mountains, the airflow climbs upward and produces plenty of precipitation. The
simulation bias is that the condensate that should have been generated in the Bay of



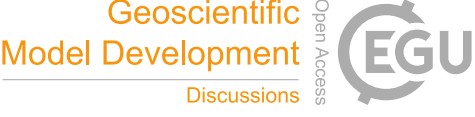

Bengal is brought to the southern slope of the Tibetan Plateau. It is noteworthy that after considering the topographic vertical velocity, the simulation results are remarkably improved. The positive precipitation deviations in the southern and eastern edges of the Tibetan Plateau and the negative deviations in the low-altitude region of the windward slope are obviously improved. Moreover, the Multi experiment (Fig. 3b) performs better than the Single experiment (Fig. 3a), and the improvement ratios of positive deviations for the Single and Multi experiments are both 20%–30% (Fig. 3d and 3e). The results above indicate that the modification of topographic vertical velocity plays a vital role in topographic precipitation simulations.

[Insert Figure 3]

More details of model performance and precipitation variations are revealed by the meridional and latitudinal averages of precipitation over the Tibetan Plateau. The meridional average precipitation though the Tibetan Plateau over 87°E–95°E (Fig. 4b) suggests that the precipitation peak for the Ctl (green line) is located north of the GPCP(black line), but more precipitation than GPCP. The precipitation distribution for the Single (blue line) experiment is the same as that for the Ctl experiment. However, the peak in Multi experiment (red line) is located north of GPCP, but the rain intensity is nearly equal. This result indicates that considering the decaying of multi-layer vertical velocity can significantly reduce the overestimation of precipitation over the south foot of the Tibetan Plateau. Fig.4a shows the latitudinal average of precipitation over 22°N–25°N. Compared with the GPCP (black line), the Ctl experiment (green line) considerably underestimates the rainfall in front of the southern edge of the Tibetan Plateau. At the eastern peak 91°E, the difference between Ctl and GPCP is about -8.41mm day$^{-1}$, and the maximum value of Multi experiment (14.51 mm day$^{-1}$) presents similar magnitude to that of GPCP (17 mm day$^{-1}$). At the windward peak 26°N, the difference between Ctl and GPCP is about 12.5 mm day$^{-1}$, and the value of Single experiment (14.1 mm day$^{-1}$) presents similar magnitude to that of GPCP (14.22 mm day$^{-1}$).

[Insert Figure 4]

To further investigate the impact of vertical circulation on precipitation simulations, Figure 5 displays the vertical pressure velocity, meridional vertical circulation and their difference averaged over 87°E–95°E. It can be found that for the Single, Multi and Ctl experiments (Fig. 5a–5c), there is strong southerly wind near 27°N–38°N, but the Ctl experiment does not simulate the variability of the vertical velocity. The vertical motion for the Single and Multi experiments appears at 28°N, which is an essential factor of orographic precipitation. Fig. 5d and 5e visually show the differences between the vertical pressure velocity and meridional-vertical circulation among the Single, Multi and Ctl experiments. Compared with Ctl experiment, the mountain blocking for the affects the Indian summer monsoon, weakening the



southerly wind component. Due to the stronger vertical motion, the vertical and southerly wind components for the Multi experiment are stronger than those for the Single experiment.

[Insert Figure 5]

In terms of the biases of model simulations, Fig. 6 presents differences in convective precipitation, large-scale precipitation, shallow convective precipitation and ZM convective precipitation between the simulations and GPCP. The deviations in the convective precipitation present almost the same spatial pattern (Fig. 6a and 6e) as the total precipitation (Figs. 3a–3b), especially along the southern and eastern edges of the Tibetan Plateau. The deviation in the spatial pattern of large-scale precipitation is slightly different (Fig. 6b and 6f). The Single and Multi experiments only revise the positive deviations of precipitation in the middle region of the southern slope (28°N–32°N, 82°E –88°E), and the simulations of Multi experiment are slightly higher than those from the Single experiment. However, both Single and Multi experiments greatly improve the negative deviations of precipitation in front of the southern slope (22°N–25°N, 90°E–97°E). The deviations in the spatial pattern of shallow convective precipitation (Fig. 6c and 6g) show almost the same between Single and Ctl experiments and between Multi and Ctl experiments, and the most negative deviations are both located at altitudes above 500 m. In the regions with altitudes below 500 m, the deviation of the ZM convective precipitation (Fig. 6d and 6h) presents almost the same spatial pattern as that of the convective precipitation (Fig. 6a and 6e).

[Insert Figure 6]

To further analyze which type of precipitation improvement is dominant, we investigate the contributions of convective precipitation, large-scale precipitation, ZM convective precipitation and shallow convective precipitation to the improvement of total precipitation simulations (Fig. 7). The results suggest that for the improvement of the overestimation of total precipitation at altitudes from 500m to 4000 m (pink shaded areas in Fig. 3c), the Multi experiment performs better than the Single experiment. The total precipitation overestimation of 12.9 mm day$^{-1}$ is improved by 6.23 mm day$^{-1}$ for the Multi experiment and 3.23 mm day$^{-1}$ for the Single experiment (Fig. 7a). For the improvement of the total precipitation simulations in the Multi experiment, the improvement of convective precipitation (4.83 mm day$^{-1}$) accounts for the largest proportion, while the improvement of large-scale precipitation is only 1.4 mm day$^{-1}$. This is due to the fact that the water vapor is lifted higher by the topographic vertical motion in the Multi experiment, which is favorable for triggering convective precipitation. In terms of convective precipitation, there is little difference in the improvement between the shallow convective and ZM convective precipitation, and the improvements of precipitation simulations are both about 2 mm day$^{-1}$. The improvement of precipitation simulation for the Single experiment is similar to that



for the Multi experiment, but the large-scale precipitation negatively contributes to the improvement of total precipitation in the Single experiment. Below 500 m, the underestimation of the total precipitation is about 3 mm day$^{-1}$, and the Single and Multi experiments both improve ~1.2 mm day$^{-1}$, but the composition of precipitation types contributing to the improvement is different (Fig. 7b). In the Single experiment, the decrease of biases comes mainly from the improvement of large-scale precipitation simulation, and the improvement of convective precipitation can be negligible. This is because in the Single experiment, the water vapor of the whole layer is lifted, and therefore the improvement of total precipitation simulation is dominated by the improvement of large-scale precipitation simulation. However, the contribution of convective precipitation to the improvement of total precipitation simulation is greater than that of the large-scale precipitation in the Multi experiment. Moreover, ZM convective precipitation is the dominant precipitation type in convective precipitation, and shallow convective precipitation makes a negative contribution to the improvement of total precipitation simulation.

[Insert Figure 7]

Since the differences in the total precipitable water (TPW) and 10m wind are related to precipitation, we analyze the distributions of the spatial differences of the 10m wind and TPW for the Single, Multi and Ctl experiments over the Tibetan Plateau (Fig. 8). Compared with Ctl experiment, the TPW shows negative deviations in the southern and eastern edges of the Tibetan Plateau in both the Single and Multi experiments. In front of the southern slope (windward), the TPW presents positive deviations in the Multi experiment (Fig. 8a) but negative deviations in the Single (Fig. 8b), indicating that the Multi experiment improves the precipitation simulation in front of the windward slope and allows the water vapor transported to the front of the southern slope of the Tibetan Plateau with the Asian monsoon. This result is consistent with the precipitation distribution in Fig. 3. Also, the 10m wind can prove this result. In the Single and Multi experiments, the wind speed in high altitude regions decreases. However, only in the Multi experiment, there are positive deviations at the southern foot of the Tibetan Plateau, i.e., low-altitude windward-slope regions (Fig. 8a–8b).

[Insert Figure 8]

Water vapor transport is a critical factor in determining precipitation distribution and an essential quantity for the orographic precipitation is the horizontal water vapor flux. As shown in Fig. 9, the water vapor transported from the northern Indian Ocean reaches the coast of the Asian continent along the Indian peninsula and the Bay of Bengal in the Ctl (Fig. 9c), Single (Fig. 9a), Multi (Fig. 9b) experiments and ERA5 (Fig. 9d). After that, the water vapor is separated into two branches, one of which reaches the southern slope of the Tibetan Plateau and flows eastward after being blocked by the plateau. The other branch transports eastward. Compared with the Ctl



experiment, more water vapor is transported from the northern Indian Ocean in the Multi experiment, and more water vapor converges in front of the southern slope of the Tibetan Plateau (80°E–87°E, 24°N–26°N), but less water vapor climbs the slope. Additionally, the water vapor transported eastward weakens due to the blocking of the plateau, forming a weakened "water belt". It can be explained by Yu et al. (2015), i.e., the altitude of land surface jumps from lower than 200 m to more than 4000 m within approximately 4 model grids, and the CAM5 (Ctl experiment) allows the multi-grid transport and spurious accumulated water vapor at cold and high-altitude regions. In contrast, the scheme of multi-layer topographic vertical motion implemented in the Multi experiment considers the climbing and bypassing of airflow. Thus, in the Multi experiment, water vapor is more in low-altitude regions and less in high-altitude regions. As a result, the precipitation is more in front of the slope and less in the southern slope of the Tibetan Plateau, which is consistent with the previous conclusion of total precipitation (Fig. 3). When the water vapor transports northward, there is a branch of water vapor in East Asia, which moves northwestward after bypassing westward and weakens markedly. This leads to a decrease in precipitation on the eastern edge of the Tibetan Plateau. Therefore, the differences between the simulations and observations, the excessive precipitation on higher slopes and less precipitation on lower slopes are considerably improved. In terms of the Single experiment, the variation of water vapor presents almost the same spatial pattern as that in the Multi experiment but less than in the Multi experiment. The only difference is that there is no noticeable increase in water vapor in lower slopes due to less pronounced variation in precipitation. Rahimi et al. (2019) investigated the relationship between the location of precipitation peak along slopes and horizontal resolution, and they found that finer resolution could allow the peak location to move northward. Previous studies found that the orographic drag of complex topography may only be resolved at horizontal resolutions of a few kilometers or even finer resolutions (Sandu et al., 2016; Wang and Zhang, 2020). However, our research demonstrates that considering the sub-grid parameterization scheme of slope gradient and surface and adding the topographic vertical motion in the CAM5-SE can address the impacts of topographic complexity on precipitation. It significantly improves the underestimation of precipitation over the windward slope of the Tibetan Plateau and the overestimation of precipitation over the steep edge of high mountains at the horizontal resolutions of hundred kilometers, which is equivalent to the horizontal resolutions of a few kilometers or the integration step of a few months in climate models (Li et al. 2022).

[Insert Figure 9]

Upslope flow is critical for orographic precipitation, which allows air to climb over mountains more easily (Smith 2019). Figure 10 presents the meridional-vertical cross-section of water vapor transport along 90°E. The results suggest that for the Single and Multi experiments (Fig. 10a and 10b), the vertical water vapor transport considerably enhances from 27°N, and even the lifting height in the Multi experiment


is higher than that in the Single experiment. Compared with the Ctl experiment, the lifting height of water vapor reaches about 700 hPa in the Single experiment (Fig. 10d), while it reaches about 650 hPa in the Multi experiment (Fig. 10e). The upslope flow supplies the water vapor to the windward slope, and the airflow blocking reduces the precipitation over the region above 500 m.

[Insert Figure 10]

A similar precipitation response can be found in other high mountains, such as the Andes in South America. Figure 11 shows the biases of precipitation simulated in the Single, Multi and Ctl experiments in South America during austral summer (December to February). It can be found that in December–February, there is strong
southerly wind at 850 hPa (Figs. 11a–11b) on the western edge of the Andes (from west of 30°S to 10°S), and large positive precipitation biases can be found in front of the foot of the Andes (Fig. 11c). In the Ctl experiment, the precipitation is overestimated on ridges above 1000 m and is underestimated in some low-altitude regions on the eastern slope. These biases are closely associated with the strong wind
at 850 hPa on the eastern edge of the Andes. In both Single and Multi experiments (Figs. 11a and 11b), the overestimation of precipitation decreases on ridges above 1000 m and increases in the windward slope at the eastern region of the Andes.

                                [Insert Figure 11]

The distributions of spatial differences in the specific humidity and TPW in South
America for the Single, Multi and Ctl experiments are shown in Fig. 12. Similar to on the Tibetan Plateau, compared with the Ctl experiment, the TPW shows negative deviations in mountain tops in both the Single and Multi experiments, which is in agreement with the precipitation distribution in Fig. 11. However, the TPW on the foot of the northeastern slope (windward) only displays positive deviations in the
Multi experiment but negative deviations in the Single experiment (Fig. 12a and b). This result suggests that the Multi experiment improves the precipitation simulation in front of the windward slope, and in both the Multi and Single experiments, the water vapor is transported to the eastern slope. Thus, the TPW accumulates in this area to form large positive deviations. The results for the specific humidity (Fig.12c–12d) and
TPW are consistent. In the Single and Multi experiments, there are dry deviations in high-altitude regions. However, only in the Multi experiment, there are wet deviations at the southern foot of the Tibetan Plateau, i.e., the low-altitude windward-slope regions.

                                [Insert Figure 12]

Table 1 presents the root mean square error (RMSE) of precipitation simulations in several typical areas with complex terrain during boreal summer (figure omitted). The results indicate that in the Tibetan Plateau, Equatorial New Guinea and Indonesia



(100°E–150°E, 10°S–10°N) and South America (30°W–90°W, 60°S–5°N), the RMSE values of precipitation simulations in the sensitivity experiments are smaller than those in the Ctl experiment. For the Ctl experiment, the RMSE is the largest in the Tibetan Plateau. Almost all GCMs have large deviations in precipitation simulations on the Tibetan Plateau. Therefore, after considering the dynamic lifting of topography, the improvement of biases in this area is the most pronounced. Moreover, the improvement of the Multi experiment is better than that of the Single experiment, reaching about 29.23%, which indicates that the steeper the mountains are, the more obvious the influence of lifting condensation on multi-layer vertical velocity is. The impact of single topographic vertical motion is limited to low-altitude areas. However, in Africa, the surface is relatively flat, and the slope gradient is small. Thus, the method in this research may not be as effective.

[Insert Table 1]

## 4 Conclusions and discussion

A common bias of the AGCMs is the overestimation of orographic precipitation. One primary reason for this bias is the imperfection of the sub-grid terrain parameterization scheme. One critical reason is that the influence of topographic lifting on airflow and water vapor transport is not considered in numerical models. In this study, we investigate whether such excessive precipitation simulation can be improved by considering the topographic vertical velocity in the CAM5-SE. The results show that the simulated precipitation in steep regions is sensitive to topographic vertical velocity. In the Multi experiment, the underestimated total precipitation is remarkably improved at lower layers on steep windward slopes. However, in the Ctl experiment, there are large dry biases, and the overestimation of precipitation in high-altitude areas of steep mountains is markedly reduced. The increase of precipitation on steep windward slopes and the decrease of precipitation in high-altitude areas of mountains are mainly due to the contribution of convective precipitation, which is greater in the Multi experiment than in the Single experiment. The improvement of precipitation simulations is closely related to dynamic lifting. If the dynamic uplifting effect is not considered, every grid is flat without considering the slope gradient and slope surface. In this case, a large amount of water vapor accumulates in high-altitude areas on the top of mountains. This is partially responsible for the excessive water vapor and precipitation in high-altitude regions of steep mountains in the Ctl experiment.

Moreover, in this study, the sub-grid parameterization scheme of the topographic vertical motion performs well in precipitation simulations over complex terrains, such as the Tibetan Plateau and the Andes in South America. Moreover, the improvement of precipitation simulations for the Multi experiment is better than that for the Single experiment. As shown in Fig. 1a, with increasing numerical model resolution, the





influence of topography on multi-layer vertical velocity weakens. Therefore, it is necessary to use high-resolution numerical experiments to verify whether the dynamic lifting effect of sub-grid topography on airflow still exists.

Notably, the improvement of precipitation simulations is noticeable over the Tibetan Plateau but not in the Rocky Mountain region in North America (figure omitted). The main reason is that in the Rocky Mountain region, the wind direction is parallel to the mountain range, and the angle between the prevailing wind direction on the western side of the mountain (steep slope) and the slope surface is close to 90°. Thus, there

can be no lifting motion caused by topography. The topographic vertical motion is not only dependent on the slope gradient, but also associated with the angle between the wind direction and the slope surface. Therefore, the large amount of water vapor from the ocean cannot be transported to the mountains. In order to understand and solve these remaining problems, more numerical experiments and more detailed analyses

should be further conducted. Moreover, when we only consider the steep slope of mountains, it would greatly impact the precipitation simulation of the regional climate. Future research is also needed to investigate the possibility of applying the topographic vertical motion scheme to extreme precipitation simulation in local areas, allowing weather models to more accurately simulate extreme precipitation caused by

topography.

**Acknowledgments:** The authors would like to thank the administrator of Beijing Normal University High Performance Computing for providing the high-performance computing (HPC) environment and technical support.

**Author contributions:** YQW and LNW designed the experiments and the scope and
structure of the manuscript. YQW carried out the simulations and analyzed the results with help from JF, ZYS, QZW and HQC. All authors gave comments and contributed to the development of the paper.

**Funding:** The research work presented in this paper was supported by the Marine S&T Fund of Shandong Province for Pilot National Laboratory for Marine Science
and Technology (Qingdao) (2022QNLM010202) and the National Natural Science Foundation of China (41830536).

**Code and Data availability:** GPCP V3.0 data is available from https://doi.org/10.5067/TTO0VJF2FSYR( last access: 29 June 2022). DEM data can be found at http://usgs.gov/. The ERA5 atmospheric datasets used in this study is
available from: https://doi.org/10.24381/cds.6860a573(last access: 29 June 2022). The source code of CAM-SE5.3 is available from http://www.cesm.ucar.edu/models/cesm1.2/(last access: 20 April 2022). The dataset related to this paper is available online via Zenodo: https://doi.org/10.5281/zenodo.7256923.

**Competing interests**. The authors declare that they have no conflict of interest.

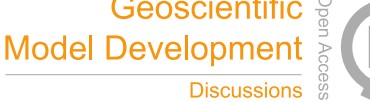

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



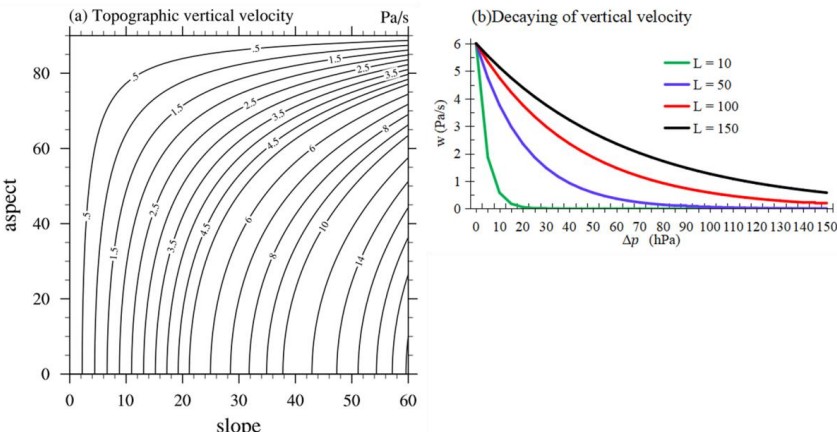

**Figure 1.** (a) Distribution of surface topographic vertical velocity (Pa/s) at different slope and aspect in 10m/s wind speed; (b) the decreasing of the unit topographic vertical velocity with height at different grid scales.


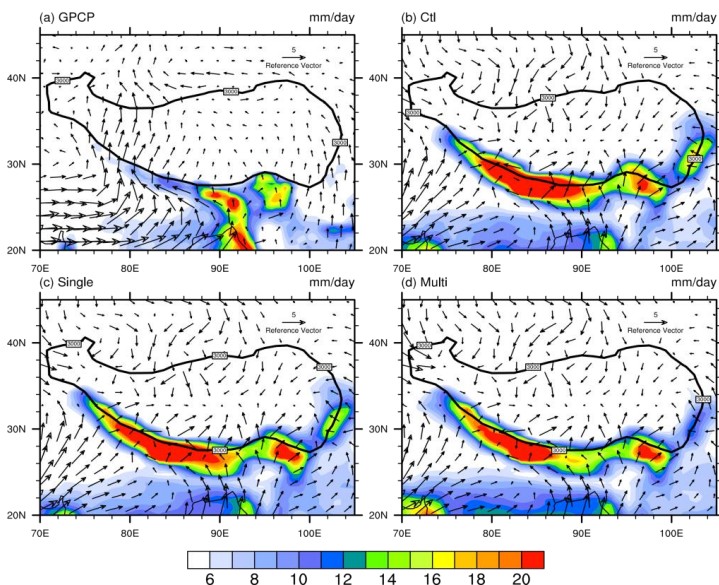

**Figure 2.** Spatial distributions of summer (June–August) average precipitation amount (mm day$^{-1}$) from (a) the GPCP data and simulation in (b) Ctl, (c) Single and (d) Multi experiments. Vectors in Fig. 2a represent the summer wind at 925 hPa, vectors in Figs. 2b–2d represent the summer wind at the lowest model level, and the black contour indicate the altitude of 3000 m.

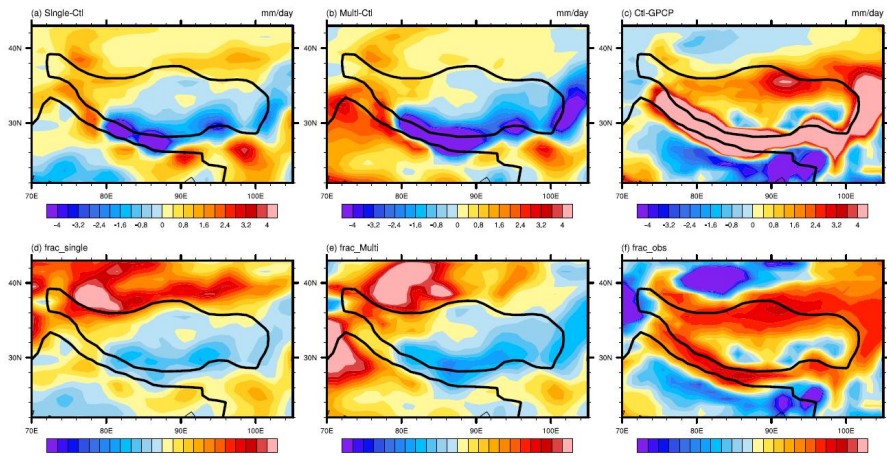

**Figure 3.** Differences of summer average precipitation amount (mm day$^{-1}$) (a) between Single and Ctl experiments, (b) between Multi and Ctl experiments and (c) between Ctl experiment and GPCP, improvement ratio of (d) Single experiment and (e) Multi experiment, (f) divergence ratio of Ctl. Black contours indicate the altitudes of 500 m and 4000 m.





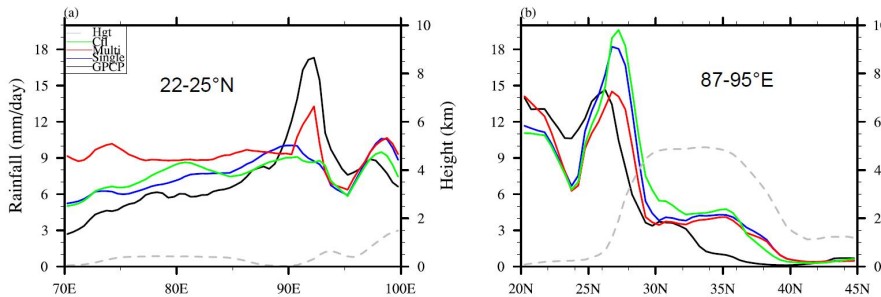

**Figure 4.** Summer precipitation averaged over (a) 22°N–25°N and (b) 87°E–95°E. Green, red, blue and black lines represent the simulated precipitation in the Ctl, Multi, Single experiments and from the GPCP data, respectively. The grey dotted lines indicate the altitudes (km).



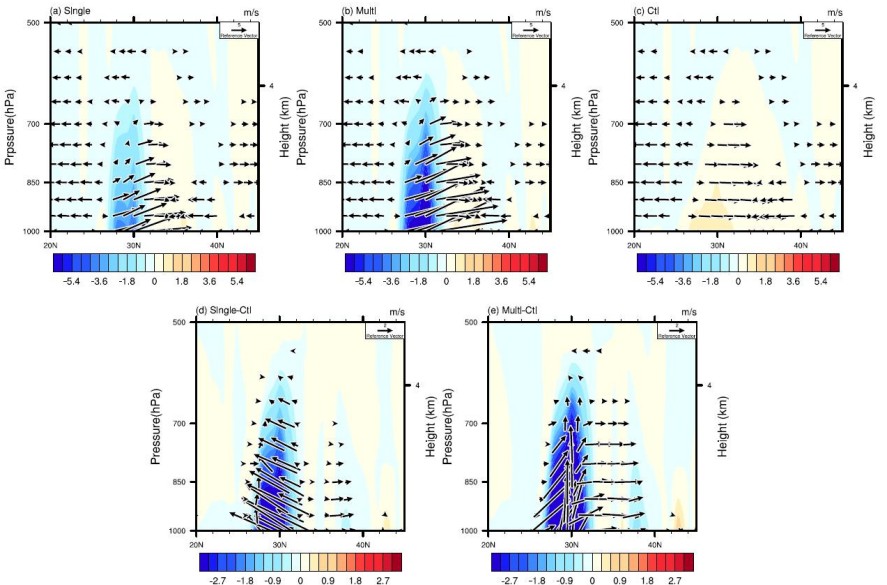

**Figure 5.** Meridional-vertical circulation (vectors) and vertical velocity (shading) averaged over 87°E–95°E in (a) Single, (b) Multi and (c) Ctl experiments, and their differences (d) between the Single and Ctl experiments and (e) between the Multi and Ctl experiments.



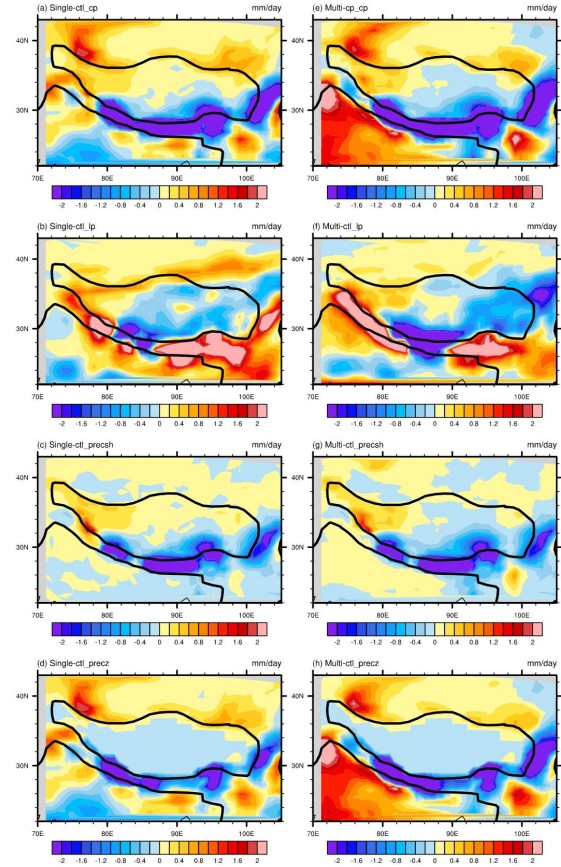

**Figure 6.** Difference of (a) convective precipitation, (b) large-scale precipitation, (c) shallow convection precipitation, (d) precipitation from ZM convection between Single and Ctl experiments. (e-h) As in (a-d) but between Multi and Ctl experiments. Black contours indicate the altitudes of 500 m and 4000 m



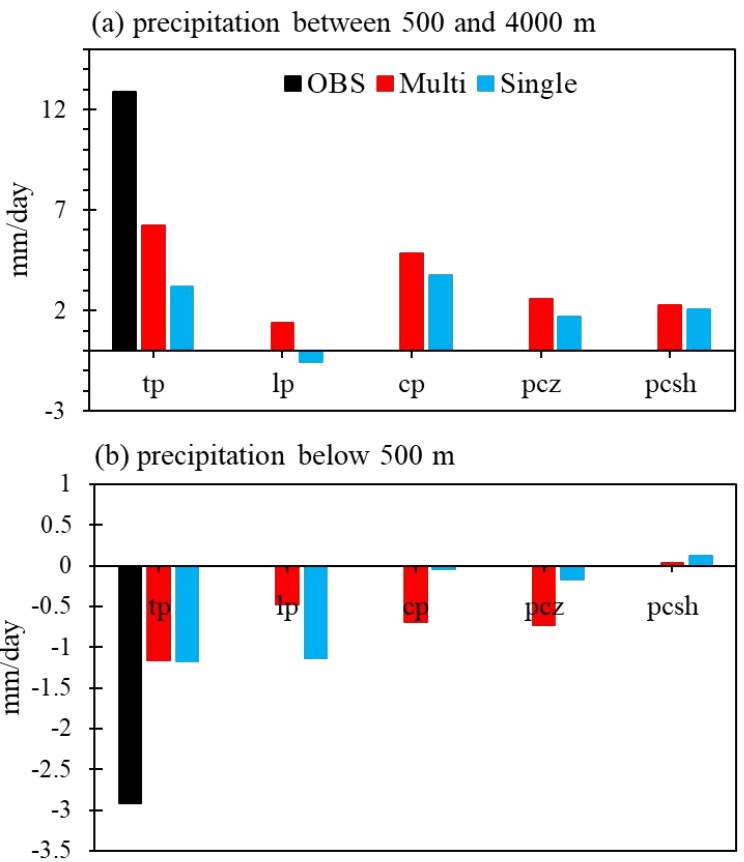

**Figure 7.** Difference of the precipitation types between the sensitivity and control experiments. (a) Positive deviations of precipitation simulations over the region with altitudes within 500–4000 m and (b) negative deviations of precipitation simulations over the region below 500 m.



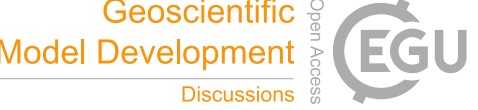
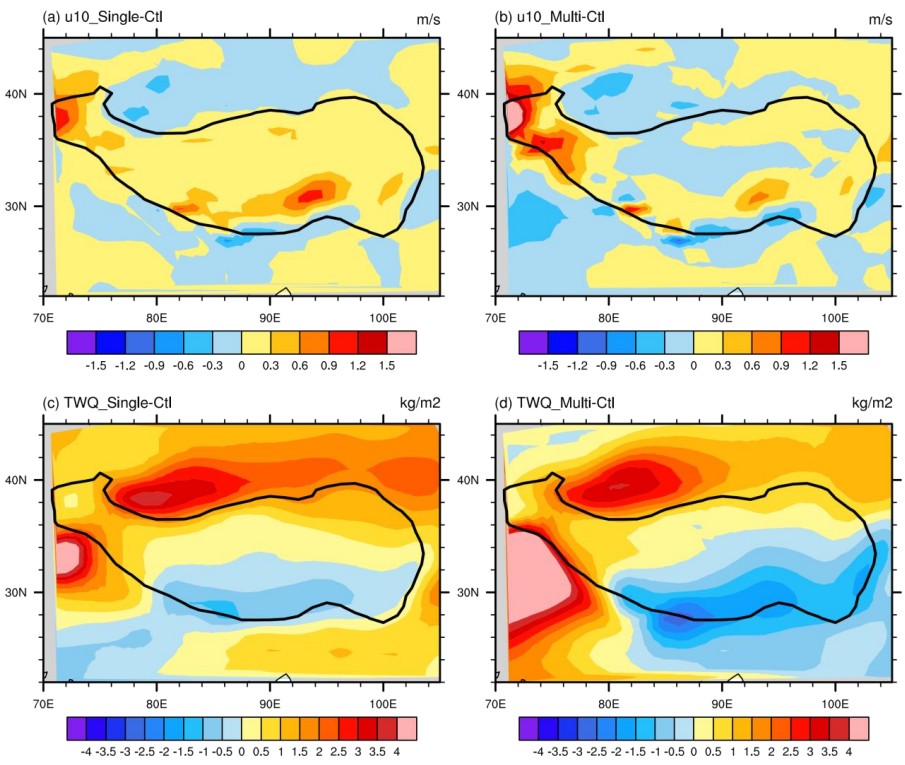

**Figure 8.** As in Fig. 6 but for (a–b) total precipitable water (kg m$^{-2}$) and (c–d) 10-m wind speed (m s$^{-1}$). Black contours indicate the altitudes of 1000 m and 2000 m.

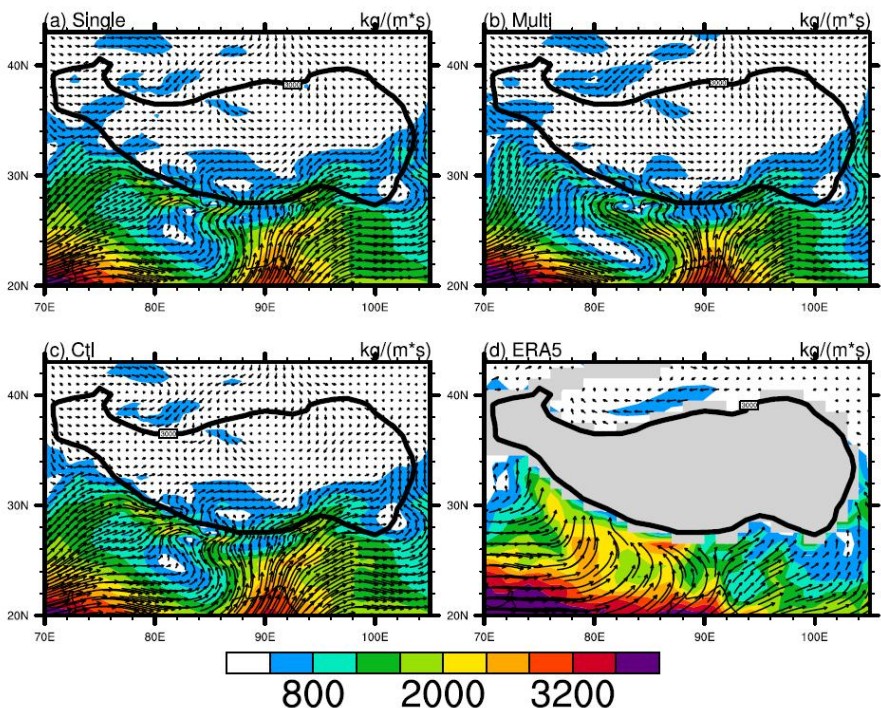

**Figure 9.** Distribution of the composite whole-layer water vapor flux (from the lowest model level to the seventh model level) in the (a) Single, (b) Multi, (c) Ctl experiments and (d) ERA5 over East Asia. Black contours indicate the altitudes of 3000 m.





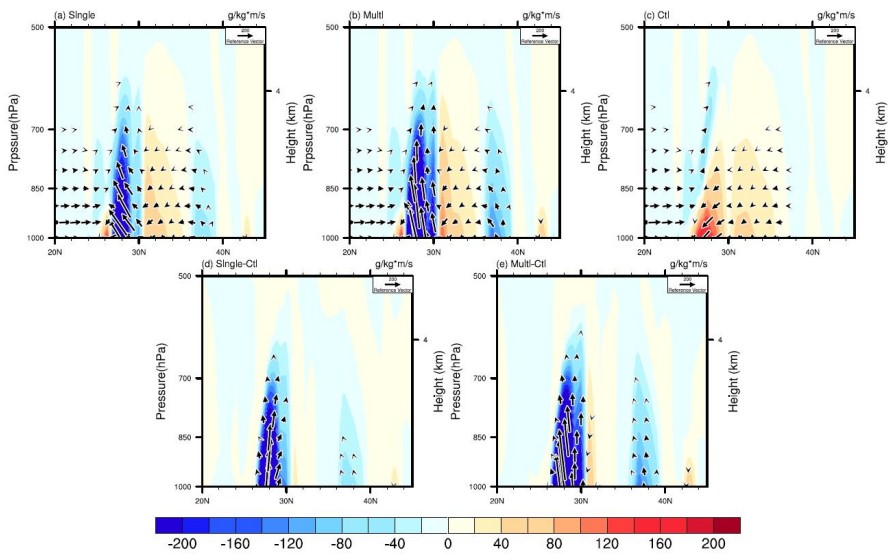

**Figure 10.** As in Fig. 5, but along 90°E.

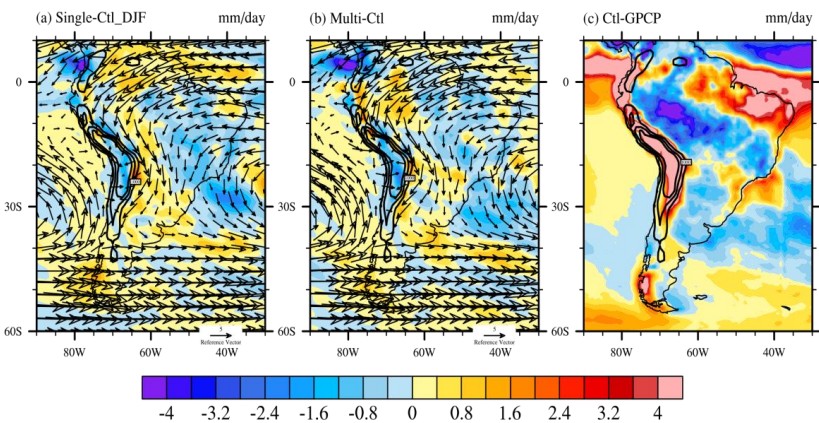

**Figure 11.** As in Fig.3a-c, but over South America. Vectors in Fig. 11a and 11b represent the 850 hPa wind in the Single and Multi experiments, respectively. Black contours indicate the altitudes of 1000 m and 2000 m.



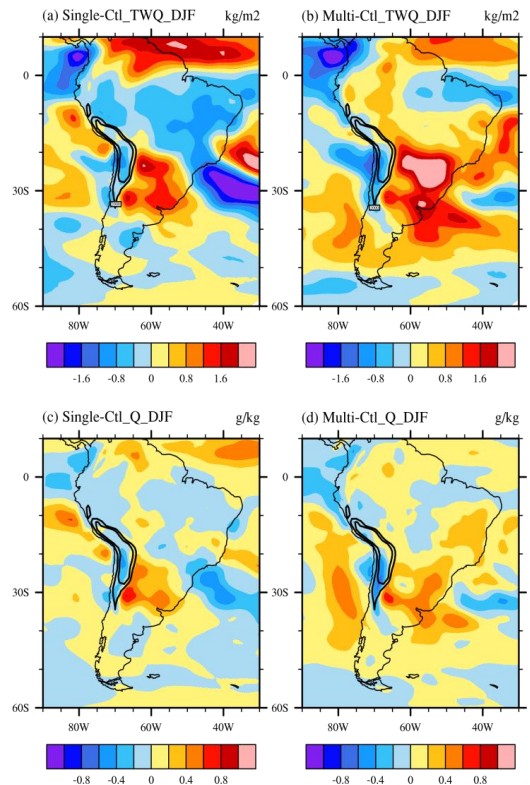

**Figure 12.** (a)-(b) Same as Fig.8(c)-(d), (c)-(d) same as (a)-(b) but for the lowest model level specific humidity over South America.

780





Table 1. RMSE in different regions.

| Regions | Ctl experiment | Single experiment | Multi experiment |
|---|---|---|---|
| Tibetan Plateau | 5.44 | 4.88 (10.3%) | 3.85 (29.23%) |
| Equatorial New Guinea | 2.55 | 2.2 (13.73%) | 1.88 (26.3%) |
| South America | 2.13 | 2.04 (4.23%) | 1.91 (10.33%) |