# Peer review of "A Sub-Grid Parameterization Scheme for Topographic Vertical Motion in CAM5-SE"

_Geoscientific Model Development, 2022_

## Author Response (AR1)

**Response to Comments of Reviewer 1**

**Manuscript number**: gmd-2022-263

**Author(s)**: Yaqi Wang, Lanning Wang, Juan Feng, Zhenya Song, Qizhong Wu, Huaqiong Cheng

**Title**: A Sub-Grid Parameterization Scheme for Topographic Vertical Motion in CAM5-SE

The article "A Sub-Grid Parameterization Scheme for Topographic Vertical Motion in CAM5-SE" presents a method / parameterisation for better representing the vertical structure of circulation and precipitation patterns at the slope of steep mountains in GCMs with relatively coarse grid. The parameterisation improved the precipitation distribution bias markedly in the model, and the authors make a good argument that bias and method may work also for many models.   On the other hand I think that the authors have a tendency to draw too general conclusions on the validity of the results both with respect to other GCMs but also for mountain slopes governed by large scale precipitation patterns rather than convection. As pointed out by the authors the improvement is mainly seen for convective parameterisation. The authors have tried to do this by also studing the Rocky Mountain region, but then just dismissed the results. While the authors point out that the method is larger for cross-mountain wind directions there are other regions, e.g. the Cascades Mountain that may have been suited for studying this. Given that this is a

development paper I do not think that this is major issues, but I still hope that the authors may include some more dicussion on the validity of the results.

We would like to express our sincere thanks for your comments. We have revised the manuscript seriously and carefully according to the reviewer's comments and suggestions. In the revision, we have tried our best to consider and incorporate all suggestions and comments. More details could be found in the revised manuscript. The detailed responses to these comments are shown below. We hope the revised manuscript could offset the shortcomings in the original manuscript. In the following, the point-by-point responses to each of the comments (**referee comments in black, our reply in blue, and revision in italics**) are attached.

In addition, the geographical distributions of boreal summer and winter mean precipitation amount from GPCP, Ctl, Single and Multi experiments over North America (155°W–122°W, 30°N–65°N) are shown in Figure R1 and R2. Due to less summer precipitation in North America and the Rocky Mountains, no analysis will be made. Although the precipitation in North America in winter is more than that in summer, the wind direction is perpendicular to the mountains, most of the improved precipitation is concentrated over the sea surface. The improvement of precipitation simulation over the Rocky Mountains the Cascades Mountain is limited. Thus, we only add the results of RMSE to Table R1, and the figures are omitted.

[Figure]

Figure R1 Spatial distributions of summer (June–August) average precipitation amount (mm day−1) from (a) the GPCP data and simulation in (b) Ctl, (c) Single and (d) Multi experiments. Vectors represent the summer wind at the lowest model level

[Figure]

Figure R2 Spatial distributions of winter (December–February) average precipitation amount (mm day−1) from (a) the GPCP data and simulation in (b) Ctl, (c) Single and (d) Multi experiments. Vectors represent the summer wind at the lowest model level

[Figure]

Figure R3 Differences of winter average precipitation amount (mm day−1) (a) between Single and Ctl experiments, (b) between Multi and Ctl experiments and (c) between Ctl experiment and GPCP, improvement ratio of (d) Single experiment and (e) Multi experiment, (f) divergence ratio of Ctl.

The paper presents the method in a detailed and organised way and I want to commend the authors on including the various mathmatical assumptions in a detailed way However when discussing the background and anlysing the results jumps in the description make the paper harder to follow. Examples of this When describing the model physics it is unclear what part is specific to the dynamical core or not; Some of the figure captions point back to other figure captions. While it is useful to know that they can be compared to previous figures so this information should

ineed be added the figures should stand on their own as well. PI think you should consider the full text to all figure captions.

Reply:Thanks to the reviewer for his/her encouragement comments. The revised manuscript has been refined according to your suggestions. These comments and suggestions greatly help us in improving the quality of this manuscript.

First, we revised the unclear description of model physics. The model mainly includes dynamic framework and physical process. "dynamical core" in our original manuscript refers to dynamic framework. It mainly includes vertical motion and horizontal motion. Our research focuses on the modification of vertical motion. We have replaced dynamic framework with dynamical core in the revised manuscript. We hope the revised figure captions could offset the shortcomings in the original manuscript.

Second, we revised the improper descriptions of the full text to all figure captions and complete the figure captions.

Final, we adjusted the organized of the manuscript. The modifications of Section 3 are as followed. 3.1 Precipitation simulation over the Tibetan Plateau: analysing the precipitation simulation and improvement over the Tibetan Plateau. 3.2 Circulation simulation over the Tibetan Plateau: analysing the horizontal and vertical circulation simulation over the Tibetan Plateau. 3.3 Precipitation simulation in other complex terrain areas: analysing the precipitation simulation over South America, New Guinea and North America. At the same time, given the RMSE over these areas and more discussions on the validity of the results. We hope that such modifications can help readers better understand our work.

Details:

1、Abstract: line 30-31. What do you mean by approaching the surface layer? top of the surface layer? Also I think that the "single -layer"multi-layer" experiments names should be introduced here since later on the use of the word single layer and multi-layer presumes that the reader see that

"In addition" introduces the split in the tests and not only another

addition.

Reply:Thank you for your valuable suggestion. The revised abstract has been

rewritten according to your suggestions. Since topography can have little effect on vertical motion when $\Delta p \geq 150hPa$, from the bottom of the model up to the layer that $\Delta p \leq 150hPa$ is called the surface layer. Approaching the surface layer refers to the vertical layers of some layers up from the lowest model layer. The surface layer in the Multi test refers to the model layer but not the boundary layer in physics. In the terrain following coordinate system, the number of layers ($\Delta p \leq 150hPa$) in different areas is different, $\Delta p$ is small in low latitude areas, which may be 6-7 layers, and $\Delta p$ is large in high latitude areas, which may be 2-3 layers, so it is dynamic layers.

**The following are revised in the revision:**

Line 30-31: We extend the dynamic lifting effect of topography from the lowest layer(Single experiment) to multiple layers, approaching the bottom model layers (Multi experiment).

2、Introduction. First paragraph: I can not see that these conclusions are

generally valid. The studies are mostly focused on tropical

and   subtropical regions as far as I can see?.

Reply:Thank you for your valuable suggestion. Tropical and subtropical precipitation

is indeed important, this is right, but precipitation is also important in the high latitudes region and produce simulation biases. The studies in the first paragraph focuses not only on precipitation simulation in the tropics and subtropics regions, but also focuses on Canada, Siberia (Liu et al., 2014), Tibetan Plateau (Cui et al., 2021) and so on. At the same time, no matter whether the region is in the tropical or subtropical region, as long as there is complex terrain, it is the key area of our research.

Liu Z, Mehran A, Phillips T J, et al.: Seasonal and regional biases in CMIP5 precipitation simulations. Clim. Res. 60(1): 35-50, 2014.

3、line 50-52. Can be condensed for readability? --> Zhu and Yang found that the model biases over the Tibetan Plateau in the CMIP6 models were even larger(more positive) than in the CMIP5 models.

Reply:Thank you for your valuable suggestion. It has been revised.

4、line 55-56 .. less precipitation in windward slopes? I do not see the conncetion

Reply:Thank you for your valuable suggestion. We do apologize for not making it clearly in the original manuscript. Navale and Singh (2020) found the bias map of the Plateau experiment shows deficient rainfall all along the plateau and a highly underestimated patch of rainfall just before the foothills of the steep slope whereas overestimated (heavy) rainfall was seen all along the slope. Because southwest wind prevails on the south side of the Tibetan Plateau in summer, before the foothills of the steep slope means windward slope. We have modified the "windward slope" and replaced it with "before the foothills of the steep slope" in the revised manuscript.

Navale A, Singh C.: Topographic sensitivity of WRF-simulated rainfall patterns over the North West Himalayan region. Atmos Res 242:105003, 2020.

**The following are revised in the revision:**

Line 55-56:Excessive precipitation was simulated in both weather/climate models and global/regional models in regions with steep and high mountains, but less precipitation before the foothills of the steep slope (Done et al., 2004; Kunz and Kottmeier, 2006; Alpert et al., 2012; Chao 2012; Navale and Singh, 2020).

5、line 71-72. Relevance? Just commenting that resolution is less important in flat areas?

Reply:Thank you for your valuable suggestion. The description of the original manuscript is not accurate enough. Just some research found that increasing spatial resolution does not always improve precipitation simulations some areas. For example, Chao et al. (2013) found that spatial density and clustering of summer

extremes in south–east England are poorly simulated in both the 12- and 1.5-km simulations. In general, we have not found any clear evidence to show that the 1.5-km simulation is superior to the 12-km simulation, or vice versa at the daily level.

**The following are revised in the revision:**

Line 71-72: However, increasing spatial resolution does not always improve precipitation simulations in some areas, for example, in lowlands of southeastern England (Chan et al. 2013; Wang et al. 2017)

6、line 95 supersaturation?

Reply:Yes, you are right, and "oversaturation" has been revised as "supersaturation" in the revision.

7、line 99-100. Suggest that you split sentence in two, remove "and only"

line 99 improve -> improved

Reply:Thanks. It has been revised.

**The following are revised in the revision:**

Line 99-102: These studies only improved the scheme of water vapor advection scheme. Shen et al. (2007) proposed a sub-grid correction parameterization scheme for pressure tendency by considering slope and orientation according to the disturbance lifting caused by each fine grid.

8、line 108 delete "in this study" ? The sentence is complex enough as it is

Reply:Thanks. It has been revised.

9、section 130-140: Suggest to rewrite to either keep all specific information at the top or at the bottom so that it is clear that the description of this part.

Reply:Thank you for your valuable suggestion. We do apologize for not making it clearly in the original manuscript. Our understanding of major model physics of CAM5-SE is not clear enough, resulting in inaccurate description. We have rewritten the description of this part.

**The following are revised in the revision:**

Line 130-140: The major model physics of CAM5-SE include: (1) the separate deep convection scheme is ZM (Zhang and McFarlane 1995; Richter and Rasch 2008). (2) The shallow convection scheme is University of Washington (UW, Park and Bretherton 2009). (3) The cloud microphysics scheme is MG1.0 (Morrison and Gettelman 2008; Gettelman et al. 2010). (4)The moist turbulence scheme for calculating sub-grid vertical transport of heat and moisture is diag_TKE (Turbulent Kinetic Energy, Bretherton and Park, 2009a). (5) The radiation scheme is Raipid Radiative Transfer Model for GCM (RRTMG) package (Mlawer et al. 1997).

10、Line 152 "rho" is a constant or is it air density (i.e. variable in time and space)

Reply:Thanks. "rho" is not a constant, it is calculated in the model codes. As you said, it is variable in time and space.

**The following are revised in the revision:**

Line 152: $\rho$ is air density and g is gravitational acceleration.

line 166 too small to be ignored ?  Does not make sense ?--> so small that it can be ignored.

Reply:Thanks for your suggestion. It has been revised.

11、line 178 T = Temperature?

Reply:Yes, T is temperature, $\theta$ is Potential temperature. According to the vertical distribution of potential temperature and the vertical atmospheric temperature reduction rate:

$$\frac{\partial \theta}{\partial z} = \frac{\theta}{T}(\gamma_d - \Gamma) \approx \frac{\theta}{T}(\gamma_v - \Gamma) \qquad (1)$$

According to static equilibrium:

$$\frac{\partial p}{\partial z} = -\rho g \qquad (2)$$

Thus,

$$\sigma = -\frac{T}{\theta}\frac{\partial \theta}{\partial p} = \frac{(\gamma_d - \Gamma)}{\rho g} \approx \frac{(\gamma_v - \Gamma)}{\rho g} \qquad (3)$$

In the calculation of topographic vertical motion, the dry adiabatic lapse rate $\gamma_d(\gamma_v)$, atmospheric vertical temperature reduction rate $\Gamma$ and the air density $\rho$ are approximately constant, $\sigma$ is also approximately a constant.

12、Section line 189-200. I think I understand what the authors try to describe but I think it can made shorter and more precise at the same time.? With L=10 km the tropographical vertical velocity is negligble 10 hPa above the surface which is lower that the top lowest CAM5 model vertical layer, so a single grid parameterisation is enough. For L=150 km the influence reaches up to 150 hPa above the surface.

Reply:Thank you for your valuable suggestion. It has been revised.

**The following are revised in the revision:**
Line 189-200: Figure 1b shows the linear variation of the unit topographic vertical velocity intensity with altitude at the given model resolution. The results indicated that

with the increase of model resolution, the topographic vertical velocity decreases rapidly with altitude. When =10km, the topographical vertical velocity is negligible 10hPa above the surface which is lower than the next layer of the lowest model vertical layer in CAM5-SE, so a single layer parameterization scheme is enough. For L=150km, the influence reaches up to 150hPa above the surface, so multi-layer topographic vertical velocity parameterization scheme is necessary. It can provide some new information for numerical simulations. Notably, preprocessing the sub-grid topographic data before the model integration may increase a small amount of computation compared with CAM5-SE.

13、line 228-230. Avoid spin-up? .. first year of simulation is discarded as spin-up

Reply:Thanks for your valuable suggestion. It has been revised.

**The following are revised in the revision:**
Line 228-230:  All the three cases are carried out for 6 years, and the first year of simulation is discarded as spin-up.

14、line 234 I presume the ERA is used for comparison with model (the sentence does not say)

Reply:Thank you for your valuable suggestion. You are right and we have revised in the manuscript.

**The following are revised in the revision:**

Line 234-238: Monthly mean atmospheric data, comprising surface pressure, specific humidity, zonal and meridional wind ((at 11 vertical levels from 1000 to 700 hPa) during 1991–2021, are from the European Centre for Medium-Range Weather Forecasts Reanalysis 5 data set (ERA5) on a 0.25° × 0.25° grid used for comparison with model results (Hans et al., 2020). And the lowest model layer wind is derived from the ERA-Interim at a 0.25° horizontal grid spacing and 60 model levels.

15、Figure 2: 2a 925 hPa wind = ERA ? Also it can not be 925 hPa given the height of Tibet, ground following level ?

Reply:Thank you for your valuable suggestion. We have revised the data and Figure in the revision. We have replaced the 925hPa wind with the lowest model level wind in Figure R4.

[Figure]

Figure R4. Spatial distributions of summer (June–August) average precipitation amount (mm day−1) from (a) the GPCP data and simulation in (b) Ctl, (c) Single and (d) Multi experiments. Vectors in Fig. R4a represent the summer wind at the lowest model level in ERA-Interim, vectors in Figs. R4b–d represent the summer wind simulation at the lowest model level, and the black contour indicate the altitude of 3000 m.

16、line 326. I do not understand this sentence. Missing words after the in the  "mountain blocking for the "

Reply:Thanks. It has been revised.

**The following are revised in the revision:**

Line326-328: Mountain blocking has an impact on the Indian summer monsoon, reducing the southerly wind component in Single and Multi experiments compared to Ctl experiment.

17、line 359. Improvement mentioned twice. Suggest shortening the sentence to: For the Multi experiment, the ... Also I think "proportion" should be replaced by "part" in the same sentence.

Reply:Thank you for your valuable suggestion. It has been revised.

**The following are revised in the revision:**

Line359-361: For the Multi experiment, the improvement of convective precipitation (4.83 mm day−1) accounts for the largest part, while the large-scale precipitation is only 1.4 mm day−1.

18、Generally on numbers and improvement/degradation in section page 11-12   Although you only have 4 degrees of freedom (5 years) I still think you should test significance.

Reply:Thanks for your suggestion. This paper focuses on improving the model performance in simulating precipitation and circulation in all regions with complex terrain after adding the topographic vertical motion schemes. As long as the sensitivity tests achieve this purpose, it is proved that the scheme is effective no matter whether has significant improvement/degradation or not, so significance is not necessary. Therefore, according to your suggestion, we have added test significance in Figure 6 and Figure8 but not analyzed in the revised manuscript.

[Figure]

Figure R5 Difference of (a–b) total precipitable water (kg m−2) and (c–d) 10-m wind speed (m s−1) between Single, Multi and Ctl experiments. Dotted areas are statistically significant at the 90% confidence level

[Figure]

Figure R6 Difference of (a) convective precipitation, (b) large-scale precipitation, (c) shallow convection precipitation, (d) precipitation from ZM convection between Single and Ctl experiments. (e-h) As in (a-d) but between Multi and Ctl experiments. Black contours indicate the altitudes of 500 m and 4000 m. Dotted areas are statistically significant at the 90% confidence level

19、I think figure 7 is somewhat confusing with a mixing of model and model -observational differences as well as different order compared to all other figures (multi before single). I think you can remove the black bar and switch single and multi. The bias of control can can be given as number in the figure text.

Reply:Thank you for your valuable suggestion. Figure7 has been revised. We have removed the black bar and switch Single and Multi.

[Figure]

Figure R7 Difference of the precipitation types between the sensitivity and control experiments. (a) Positive deviations of precipitation simulations over the region with altitudes within 500–4000 m and (b) negative deviations of precipitation simulations over the region below 500 m.

20、Figure 8 a-b and c-d are switched compared to the figure caption. The article points to a and b for both TPW and wind so must also be corrected.

Reply:Thank you for your valuable suggestion. We have corrected the figure.

[Figure]

Figure R8 Difference of (a–b) total precipitable water (kg m$^{-2}$) and (c–d) 10-m wind speed (m s$^{-1}$) between Single, Multi and Ctl experiments. Dotted areas are statistically significant at the 90% confidence level

21、line 442. I do not understand the latter part of this sentence. "...the integration step of a few months"

Reply: Thanks for your suggestion. We do apologize for the description of the original manuscript is not accurate enough. "..the integration step of a few months" means simulation time, it has been revised.

**The following are revised in the revision:**

Line 438-443: It significantly improves the underestimation of precipitation over the windward slope of the Tibetan Plateau and the overestimation of precipitation over the steep edge of high mountains at the horizontal resolutions of hundred kilometers, which is equivalent to the horizontal resolutions of a few kilometers or a few months simulation in climate models (Li et al. 2022).

22、Figure 9: The ERA 5 values is from below a certain height since there are not values over Tibet.?

Reply: Yes, it is right. ERA5 data is missing over the Tibetan Plateau.

23、Table 1: Even though you have discussed the bias over Tibet several times during the article, I think you should also include bias in table 1, in particular since you do not include any bias number from the Maritime Continent, and South America in the text. I think you can also add the latitude-longitude information in table 1.

Reply:Thank you for your valuable suggestion, we have added the latitude-longitude information in Table 1 and added the bias over North America. By the way, we also added some analysis about Table 1.

**Table R1**. RMSE in different regions.

| Regions | Ctl experiment | Single experiment | Multi experiment |
|---|---|---|---|
| Tibetan Plateau (70°E–105°E, 22°N–45°N) | 5.44 | 4.88 (10.3%) | 3.85 (29.23%) |
| Equatorial New Guinea (100°E–150°E, 10°S–10°N) | 2.55 | 2.2 (13.73%) | 1.88 (26.3%) |
| South America (30°W–90°W, 60°S–5°N) | 2.13 | 2.04 (4.23%) | 1.91 (10.33%) |
| North America (155°W–122°W, 30°N–65°N)) | 1.57 | 1.46 (7%) | 1.42(9.55%) |

**The following are revised in the revision:**

Line 485:499: Table 1 presents the root mean square error (RMSE) of precipitation simulations in several typical areas with complex terrain during boreal summer or winter (figure omitted). The results indicate that in the Tibetan Plateau (70°E–105°E, 22°N–45°N, boreal summer precipitation), Equatorial New Guinea and Indonesia (100°E–150°E, 10°S–10°N, boreal summer precipitation), South America (30°W–90°W, 60°S–5°N, boreal winter precipitation), and North America (155°W–122°W, 30°N–65°N, boreal winter precipitation) the RMSE values of precipitation simulations in the sensitivity experiments are smaller than those in the Ctl experiment. For the Ctl experiment, the RMSE is the largest over Tibetan Plateau (5.44) and the smallest over North America (1.57). Almost all GCMs have large deviations in precipitation

simulations on the Tibetan Plateau. Therefore, after considering the dynamic lifting of topography, the improvement of biases in this area is the most pronounced, followed by Equatorial New Guinea (26.3%) and the smallest in North America (9.55%). Moreover, the improvement of the Multi experiment is better than that of the Single experiment, reaching about 29.23%, which indicates that the steeper the mountains are, the more obvious the influence of lifting condensation on multi-layer vertical velocity is. The impact of single topographic vertical motion is limited to low-altitude areas. However, in Africa, the surface is relatively flat, and the slope gradient is small (Wang et al., 2022). Thus, the method in this research may not be as effective so it is no longer mentioned in Table 1.

Reference:

Wang, Y., Wang, L., Feng, J. et al.: A statistical description method of global sub-grid topography for numerical models. Clim Dyn. https://doi.org/10.1007/s00382-022-06447-2,2022.

24、Line 497: This is the only time Africa is mentioned. Should be South America ? BTW I do think that the steepness over South America is quite high, but with no platau.

Reply:Thanks for your suggestion. We do apologize for unclear description due to lack of context and references but we really want to express Africa here. We have added some information in the revised manuscript.

**The following are revised in the revision:**

However, in Africa, the surface is relatively flat, and the slope gradient is small (Wang et al., 2022). Thus, the method in this research may not be as effective so it is no longer mentioned in Table 1.

Reference:

Wang, Y., Wang, L., Feng, J. et al.: A statistical description method of global sub-grid topography for numerical models. Clim Dyn. https://doi.org/10.1007/s00382-022-06447-2,2022.

25、Section 4: I can not see much discussion here   the word "discussions". I think the word can written together with results, as results and discussion.

Reply:Thank you for your valuable suggestion. We have adjusted the organized of the manuscript according to your comment. Section 4 has been revised as conclusion, we delate "discussion".

26、Line 511: I do not understand the sentence. Do you compare multi to ctl,?

Reply:Thanks for your suggestion. We are so sorry that you can't understand this sentence because of our mistakes. It is true that we compare Multi to Ctl and we have revised this sentence.

**The following are revised in the revision:**

Line 511-512: However, in the Ctl experiment, there are large dry biases, but the overestimation of precipitation in high-altitude areas of steep mountains is markedly reduced in Multi experiment.

27、Line 532. While it is fine to omitt figures I think you should either moved the discussion on Rocky Mountains to the results part or delete it altogether.   If you keep it I also suggest that you include numbers for bias and rmse in table 1. As mentioned earlier I do not think you have chosen the relevant part of North America for your analysis, but I do think it is critical for the paper.

Reply:Thanks for your suggestion. We have added North America in Table 1 and added some analysis in results.  And we have adjusted the organized of the manuscript according to your comments. The modifications of Section 3 are as followed. 3.1 Precipitation simulation over the Tibetan Plateau: analysing the precipitation

simulation and improvement over the Tibetan Plateau. 3.2 Circulation simulation over the Tibetan Plateau: analysing the horizontal and vertical circulation simulation over the Tibetan Plateau. 3.3 Precipitation simulation in other complex terrain areas: analysing the precipitation simulation over South America, New Guinea and North America.

**Response to Comments of Reviewer 2**

The results are promising, but the description of the parameterization is completely missing. The authors describe a very reasonable way to approximate \omega_s near the surface, based on the standard free-slip boundary condition at the terrain-following surface. They also give the standard diagnostic equation used to compute \omega in pressure coordinate models. They then decomposed \omega (the value computed by the model using Eq 3) into \omega_0 and \omega_s, with \omega_0 defined by equation 4.  How is this decomposition used? Just knowing the decomposition of \omega into these two parts will not of course change any results.  Then there is a comment on line 169 that I interpret to be claiming that \omega in CAM6 is only used at the surface layer and ignored in the rest of the layers. Is this really true? If this is true, then the authors must have modified some of the parameterizations (especially those contributing to convective precip) so that they do depend on \omega or \omega_s? Which parameterizations?  how where they changed? However, I believe in CAM6, the CLUBB parameterization does make use of \omega, and there may be other parameterizations. Are the parameterizations that do use \omega modifed to make use of \omega_s instead?  What about at model layers far from the surface? In addition, \omega is also used in the CAM-SE dycore, as it appears in the tendency term in the dycore.  Is this \omega replaced by \omega_s or modified in

anyway by \omega_s? Some details are given in lines 201-215, but this description is again only focused on computing \omega_s and does not describe how \omega_s could be used to change the parameterized precipitation. Also, this description is very unclear to me. I suppose it is better described in Wang et al. 2022, but the paper under review could use a high level summary of the algorithm. Up to this point, I was assuming the authors would compute \omega_s by taking the dot product of the model's prognosed surface winds with the gradient of the model's topography. But here they seem to be using a subgrid calculation computed from a high resolution topography data set (not the topography used by the model) and it's not clear how this subgrid information will be used in the model. Due to my inability to understand what the authors changed in the model, I'm not going to review the results section of the paper and I limit my comments to section 1-2.

**Reply:** We would like to express our sincere thanks for your valuable comments. We do apologize for missing the description of how /omega affects the dynamic framework and how the sub-grid topographic parameterization acts on /omega_s that misunderstand you. These comments and suggestions greatly help us in improving the quality of this manuscript. We have tried our best to consider and incorporate all suggestions and comments. We added detailed descriptions of the effect of vertical velocity on dynamic core and the parameterization scheme in the revised manuscript (Lines 131-139). And the model used in our manuscript is CAM5 but not CAM6.

Given description of the coordinate in CAM5-SE, the continuous system of equations can be written following the first law of thermodynamics, Kasahara (1974) and Simmons and Strüfing (1981). The prognostic equations are as followed (Neale et al., 2010).

$$\frac{\partial \zeta}{\partial t} = \mathbf{k} \cdot \nabla \times \left(\frac{\mathbf{n}}{\cos \phi}\right) + F_{\zeta_H},$$

$$\frac{\partial \delta}{\partial t} = \nabla \cdot \left(\frac{\mathbf{n}}{\cos \phi}\right) - \nabla^2 (E + \Phi) + F_{\delta_H}$$

$$\frac{\partial T}{\partial t} = \frac{-1}{a \cos^2 \phi}\left[\frac{\partial}{\partial \lambda}(UT) + \cos \phi \frac{\partial}{\partial \phi}(VT)\right] + T\delta - \dot{\eta}\frac{\partial T}{\partial \eta} + \frac{R}{c_p^*}T_v\frac{\omega}{p}$$

$$\qquad + Q + F_{T_H} + F_{F_H}$$

$$\frac{\partial q}{\partial t} = \frac{-1}{a \cos^2 \phi}\left[\frac{\partial}{\partial \lambda}(Uq) + \cos \phi \frac{\partial}{\partial \phi}(Vq)\right] + q\delta - \dot{\eta}\frac{\partial q}{\partial \eta} + S$$

$$\frac{\partial \pi}{\partial t} = \int_1^{\eta_t} \nabla \cdot \left(\frac{\partial p}{\partial \eta}\mathbf{V}\right) d\eta$$

(1)

The third equation in Eqs1 above shows that in the SE dynamic framework, vertical velocity affects the tendency of temperature $\frac{\partial T}{\partial t}$ directly, and also affects pressure P through the equation of state $P = \rho RT$ indirectly. And then the correction of vertical velocity can change the atmospheric circulation and precipitation.

We dose not decompose \omega into omega0 and omega_s. We give the Eq4 to show that vertical velocity (/omega) should be composed of omega0 and omega_s in topographic area, where omega_s is the topographic vertical velocity related to sub-grid slope and aspect. But CAM5-SE do not take omega_s into consideration, only consider omega0. Thus, we added omega_s into CAM5-SE according to Eq5 in the original manuscript.

Line 169 means some research considered omega_s of the lowest model layer in numerical models. That is Shen et al. (2007) proposed a sub-grid correction parameterization scheme for pressure tendency in hybrid sigma coordinate model of Nanjing University but not in CAM5-SE.

We do not use the parameterization scheme on omega, but on topography. We proposed a sub-grid topographic parameterization scheme of slope and aspect for omega_s. The topography data used in this study is from the United States Geological Survey (USGS) Digital Elevation Model (DEM) with a resolution of 1 km×1 km (Sub Grid). The simulations are performed at the horizontal resolution of ne30 (about 1°, Coarse Grid). Thus, the coarse grid contains several sub grids. We define a coarse grid as a terrestrial grid when the number of sub-grids on land occupies more than 10% of the total number of sub-grids, otherwise it is a marine grid. If the number of sub-grids with slope ≥ 5° in the terrestrial grid exceeds 10%, the terrestrial grid is considered as a complex topographic area coarse grid and needs to be parameterized. After that, the product of the trigonometric functions of the slope and aspect of each sub-grid in complex topographic area coarse grid is calculated, that is $\tan\theta_N \times \cos\varphi_N$ (TC) and $\tan\theta_N \times \sin\varphi_N$ (TS) . According to Wang et al. (2022), it was found that the sub grids contained in the coarse grids of all topographic areas follow Gaussian distribution. Then the representative value of several sub-grid topography values at the coarse grid scale is selected ($y_p = \mu + Z_p * \sigma$) and can be easily described and applied (Wang et al., 2022). Finally, bring the representative value into Eq5 in the original manuscript to

calculate omega_s.

Furthermore, we did not modify the physical parameterization schemes related to precipitation, only made modifications of the vertical velocity in CAM5-SE by adding omega_s, which is the topographic vertical motion due to sub-grid topographic dynamic uplift.

Reference:

Kasahara A (1974) Various vertical coordinate systems used for numerical weather prediction, Mon. Wea. Rev., 102, 509–522.

Simmons AJ and Strüfing R (1981) An energy and angular-momentum conserving finite difference scheme, hybrid coordinates and medium-range weather prediction, Technical Report ECMWF Report No. 28, European Centre for Medium–Range Weather Forecasts, Reading, U.K.,68 pp.

Neale RB, Chen C, Gettelman A, Lauritzen PH, Park S, Williamson DL, Conley AJ, Carcia R, Kinnison D, Lamarque J, Marsh D, Mills M, Smith AK, Tilmes S, Morrison H, Cameron-Smith P, Collins WD, Iacono MJ, Easter RC, Ghan SJ, Liu X, Rasch PJ, Tayloy MA (2010) Description of the NCAR community atmosphere model (CAM 5.0). NCAR tech note TN-486.

Minor comments:

line 103 "P-\sigma regional climate mode of Nanjing University" what does "P-\sigma" mean? the name of the model? or maybe saying it is a pressure based hybrid sigma coordinate model?

**Reply:** We greatly appreciate the suggestion. Because it is habitually called P-σ regional climate model in Chin, but it is actually pressure based hybrid sigma coordinate model. We have revised.

line 150: define aspect (I'm assuming its the direction (angle) of the face of the slope)

**Reply:** Yes you are right. We have added the definition of aspect in the revised manuscript.

line 164: typo: double commas

**Reply:** Thank you for your suggestion, we have removed excess commas in the revised manuscript.

line 168 "However, when the slop is less than ~5 degrees, the topographic vertical velocity is too small to be ignored"? Should this be so small that it

can be ignored?

**Reply:** Thanks for your suggestion. It has been revised.

line 145-150:   $V\_0$, $V\_s$ and V (no subscript) all appear in the section, but only $V\_s$ is defined. what are the other V's?

**Reply:** Thanks for your suggestion. $V\_0$ and $V\_s$ are the same physical quantity, they are indicated the surface wind velocity. V(no subscript) represent vector wind, not just surface wind. It is just written in different ways in different research. We have made a uniform modification in $V\_s$.

line 169:   "Generally, only the single layer, that is the lowest layer, is considered in numerical models.   however, in the actual atmospheric movement, the vertical motion not only affects the surface layer, but also affects near surface layers" What does this mean?   In the model used here for example, the dycore computes \omega at every model layer via equation 3, and this \omega is known to all physics parameterizations. Are the authors saying that all of these parameterizations dont make use of \omega?

**Reply:** Thanks for your suggestion. We do apologize for our imprecise expression may lead to misunderstanding. Here we indicated that topographic vertical velocity due to topographic forced uplift are not considered in CAM5-SE. Although some research considered omega_s of the lowest model layer in numerical models. That is Shen et al. (2007) proposed a sub-grid correction parameterization scheme for pressure tendency in hybrid sigma coordinate model of Nanjing University but not in CAM5-SE. According to our research, only this one study has considered the dynamics of sub-grid topography, but only the lowest model level. CAM5-SE only makes use of omega0 but not omega_s in Eq3. Thus, we added topographic vertical velocity omega_s in CAM5-SE.

**The following are revised in the revision:**

Line 169: Shen et al. (2007) proposed a sub-grid correction parameterization scheme for pressure tendency in hybrid sigma coordinate model of Nanjing University. However, the topographic vertical motion not only affects the lowest model level, but also affects near surface layers. Thus, we extend the topographic vertical velocity from single layer to multi layers.

Equation 7:    What is sh()?

**Reply:** Sorry, the symbol "sh()" is inappropriate here. It should be "sinh()". "sinh()" is hyperbolic sine function.

line 181:    "Taylor's formula" is usually referred to as the Taylor series.

**Reply:** Thanks for your suggestion. It has been revised.

line 199: "Notably, preprocessing the sub-grid topographic data before the model integration may increase a small amount of computation compared with CAM5-SE" what does this mean?

**Reply:** Thanks for your valuable suggestion. Our imprecise expression may lead to misunderstanding. Here we indicate that the topography data used in this study is from the United States Geological Survey (USGS) Digital Elevation Model (DEM) with a resolution of 1 km×1 km (Sub Grid). The simulations are performed at the horizontal resolution of ne30 (about 1°, Coarse Grid). Thus, the coarse grid contains several sub grids. We define a coarse grid as a terrestrial grid when the number of sub-grids on land occupies more than 10% of the total number of sub-grids, otherwise it is a marine grid. If the number of sub-grids with slope ≥ 5° in the terrestrial grid exceeds 10%, the terrestrial grid is considered as a complex topographic area coarse grid and needs to be parameterized. After that, the product of the trigonometric functions of the slope and aspect of each sub-grid in complex topographic area coarse grid is calculated, that is $\tan\theta_N \times \cos\varphi_N$(TC) and $\tan\theta_N \times \sin\varphi_N$(TS). According to Wang et al. (2022), it was found that the sub grids contained in the coarse grids of all topographic areas follow Gaussian distribution. Then the representative value of several sub-grid topography values at the coarse grid scale is selected ($y_p = \mu + Z_p * \sigma$) and can be easily described and applied (Wang et al., 2022).

Second, for a given grid resolution, $\frac{\sqrt{\sigma}}{2dl \times f \times sin(lat)}$ in each grid can be treated as a constant and can be preprocessed offline during the Equation 7.

Both of the above are available for offline preprocessing before the model integration, thus it may just increase a small amount of computation. And the air pressure P in equation 7 is changing with time and place, while the calculation of hyperbolic sine function is very complicated ($\sinh x = \frac{e^x - e^{-x}}{2}$), each calculation needs to use two power functions and four times in total. After simplifying Equation 7 to Equation 8 in the original manuscript, only one power function calculation is required. The improper use of the words "increase a small amount of computation compared with CAM5-SE" was replaced with "simplify calculation" in the revision. This section has been re-written

**The following are revised in the revision:**

Lines 198-200: Notably, preprocessing the sub-grid topographic data before the model integration may simplify calculation.

Line 215-233: The topography data used in this study is from the United States Geological Survey (USGS) DEM with a resolution of 1 km×1 km (Sub Grid). The simulations are performed at the horizontal resolution of different model grid (Coarse Grid). Thus, the coarse grid contains several sub grids. We define a coarse grid as a terrestrial grid when the number of sub-grids on land occupies more than 10% of the total number of sub-grids, otherwise it is a marine grid. If the number of sub-grids with slope ≥ 5° in the terrestrial grid exceeds 10%, the terrestrial grid is considered as a complex topographic area coarse grid and needs to be parameterized. After that, the product of the trigonometric functions of the slope and aspect of each sub-grid in complex topographic area coarse grid is calculated, that is $\tan\theta_N \times \cos\varphi_N$ (TC) and $\tan\theta_N \times \sin\varphi_N$ (TS) . According to Wang et al. (2022), it was found that the sub grids contained in the coarse grids of all topographic areas follow Gaussian distribution. Then the representative value of several sub-grid topography values at the coarse grid scale is selected ($y_p = \mu + Z_p * \sigma$) and can be easily described and applied (Wang et al., 2022). Finally, bring the representative value into Eq.(6) to calculate $\omega_s$. Before the experiments, advanced preprocessing is used to calculate the probability densities of the trigonometric function and grid weights.

---

## Referee Report (RR1)

I would like to thank the authors for their work on taking my comments and have no into account and see no need for further iterations.

A number of my comments was on why the the modifications did not have a large impact everywhere. I may have not myself clear that even if there are areas where there are no improvements which may slightly lessen the usefulness of an idea / formulation, discussing why and where this happens is very important for the scientific understanding of the process. I thank the authors for working on this.

A handful of almost cosmetic issue that likely will be caught by the publisher, but adding them here since I noticed them

line 299 missing space between "N" and "and"
line 326 and 397 missing space between "500" and "m"
line 324 missing space between 80% and "("

line 595 in Multi experiment → in the Multi experiment

---

## Author Response (AR2)

**Response to Comments of Reviewer 2**

**Manuscript number**: gmd-2022-263

**Author(s)**: Yaqi Wang, Lanning Wang, Juan Feng, Zhenya Song, Qizhong Wu, Huaqiong Cheng

**Title**: A Sub-Grid Parameterization Scheme for Topographic Vertical Motion in CAM5-SE

I appreciate the improved description of the parmaterization. It addresses several of the issues I raised in my first review. This is not a typical parameterization, as it does not appear in the physics package, but instead replaces the vertical pressure velocity "omega" used in the dynamical core. As this is slightly unconventional, it is good that it is clearly explained. Now that I understand this aspect of the approach, it raises additional questions:

1. What is omega_0 introduced in equation 5? This variable is not defined. Note that in Equation 7, we get the final form of \omega that will be used by the dynamical core. This final form depends on omega_0. Thus without knowing \omega_0, it is not possible to reproduce these results. From the author's review comments, I assume \omega_0 is the value of \omega computed by the dynamical core, from the standard diagnostic equation. But this needs to be clearly explained in the paper. And if my assumption is correct, the diagnostic equation for \omega_0 should be given along with

the other equations in Equation 2.

**Reply:** Thank you for your affirmation of the parameterization scheme. We would like to express our sincere thanks for your valuable comments. It is worth noting that we dose not replaces the vertical pressure velocity "omega" used in the dynamical core, but specifically added a term 'omega_s'. And we do apologize again for the inaccurate description of physical quantities. It is true that \omega_0 is the value of \omega computed by the dynamical core, from the standard diagnostic equation in equation 5. The diagnostic equations for \omega_0 is $\omega = Dp/Dt$ in CAM5, it has been shown in Line 130. Due to the fact that Formula 1, which we specifically discussed separately, was derived from the diagnostic equation, it was not given along with the other equations in Equation 2.

2. Continuing to assume that \omega_0 is the \omega computed by the standard diagnostic equation, then the final \omega used by the dynamical core appears to have some double counting: \omega_0 at the surface will be the model's horizontal velocity multiplied by the gradient of the model's topography. The author's then add to this \omega_s, which has their more sophisticated representation of the surface omega, including subgrid topographic details. Assuming a very smooth mountain, well resolved by the model, then it seems that \omega_s and \omega_0 would agree, and thus we have some double counting. I would think \omega_0 needs to be reduced to zero as one approaches the surface in a way similar to how \omega_s is reduced away from the surface.

**Reply:** Thanks for your suggestion. \omega_0 is the \omega computed by the standard diagnostic equation, related to dp/dt, it does not include the influence of sub-grid topography. To distinguish it from the topographic vertical velocity, we call it omega_ 0, but omega_0 is actually the omega in Eq1. If the model cannot recognize small terrain, there is no topographic vertical motion. Obviously, it does not exist in CAM5-Ctl. Assuming that a smooth mountain has a hemispherical shape, well resolved by the model with a certaon resolution, it still has a slope, and the slope is completely equal for each sub grid. While omega_ s is the sub-grid scale topographic vertical motion calculated from DEM data with a resolution of 1 km×1 km. It dose not double counting in our parameterization scheme because the uplifting caused by its terrain still exists.

3. One potential difficulty with this approach (modifying \omega in the dynamical core): How would this be implemented in a model that uses potential temperature as a prognostic variable instead of temperature? With such a set of prognostic variables, \omega does not appear in the dynamical equations, and thus how would one modify the equations so that they could feel the influence of changing \omega near the surface?

**Reply:** Thanks for your suggestion. The first law of thermodynamics:
$$dQ = C_v dT + p d\alpha \quad (R1)$$
where $\alpha$ is specific volume of dry air. According to the state equation:
$$p\alpha = RT \quad (R2)$$
Diagnostic equation:
$$\omega = \frac{dp}{dt} \quad (R3)$$
And the relationship between specific pressure heat capacity $C_p$ and specific constant capacity heat capacity $C_v$:
$$C_v + R = C_p \quad (R4)$$
thus :
$$\frac{dT}{dt} = \frac{kT}{P}\omega + \frac{Q}{C_p} \quad (R5)$$

And $\theta = T(\frac{p_0}{p})^{\wedge}k$, $k = R/C_p$ is a constant. Finally, the potential temperature as a prognostic variable instead of temperature in a model is: $\theta' = \theta\left(1 + \frac{k\omega}{p}dt\right)$(Omitting the intermediate derivation process). Thus, by modifying $\omega$, the potential temperature $\theta$ can be further changed. Transforming dT/dt into d$\theta$/dt is more complex than directly calculating temperature. This process is included in the Finite Volume Dynamical Core, but due to the fact that the potential temperature itself includes the factor of altitude, the improvement effect of changing the vertical velocity $\omega$ on precipitation is not significant. Therefore, we use Spectral Element Dynamical Core (CAM5-SE) in our manuscript.

4. Note that Equation 4 is only valid on the surface, so \omega on the left-hand-side needs an s subscript. Since Equation 5 comes from Equation 3 and Equation 4 (both only valid at the surface) it seems equation 5 is only valid at the surface. It is extended beyond the surface in equation 7. But

then there is a problem with the notation, since Equation 5 and Equation 7

both use \omega on the left hand side, but clearly these are different fields.

**Reply:** Thanks for your suggestion. We have revised the notation in Eq4 and Eq7 acccording to your comment. $\omega$ on the left-hand-side in Eq4 has been revised as $\omega_s$. It indicated the topographic vertical velocity of the lowest model layer. $\omega$ on the left-hand-side in Eq7 has been revised as $\omega_l$, it indicated the decrease of multi-layer topographic vertical velocity from lowest model layer up to the model layer that 150hPa above the lowest model layer.

**The following are revised in the revision:**

$$\omega_s = \frac{\partial p_s}{\partial t} + \vec{V}_s \cdot \nabla Z_s, \qquad (4)$$

$$\omega = \omega_0 + \omega_s = \frac{dp}{dt} + \omega_s \qquad (5)$$

$$\omega_l = \frac{dp_l}{dt} + \omega_s \times \gamma, \qquad (7)$$

Where $\omega_l$ is multi-layers topographic vertical velocity, $p_l$ is multi-layers pressure. $\gamma$ indicates the attenuation coefficient of topographic vertical velocity $\omega_s$ and it increases with the elevation.